# Continuous Fentanyl Infusion in Newborns with Hypoxic–Ischemic Encephalopathy Treated with Therapeutic Hypothermia: Background, Aims, and Study Protocol for Time-Concentration Profiles

**DOI:** 10.3390/biomedicines11092395

**Published:** 2023-08-27

**Authors:** Licia Lugli, Elisabetta Garetti, Bianca Maria Goffredo, Francesco Candia, Sara Crestani, Caterina Spada, Isotta Guidotti, Luca Bedetti, Francesca Miselli, Elisa Muttini Della Casa, Maria Federica Roversi, Raffaele Simeoli, Sara Cairoli, Daniele Merazzi, Paola Lago, Lorenzo Iughetti, Alberto Berardi

**Affiliations:** 1Neonatal Intensive Care Unit, Women’s and Children’s Health Department, University Hospital of Modena, 41100 Modena, Italy; garetti.elisabetta@aou.mo.it (E.G.); guidotti.isotta@aou.mo.it (I.G.); bedetti.luca@aou.mo.it (L.B.); dellacasa.elisa@aou.mo.it (E.M.D.C.); roversi.federica@aou.mo.it (M.F.R.); alberto.berardi@unimore.it (A.B.); 2Division of Metabolic Diseases and Drug Biology, Bambino Gesù Children’s Hospital, Scientific Institute for hospitalization and care (IRCCS), 00100 Rome, Italy; biancamaria.goffredo@opbg.net (B.M.G.); raffaele.simeoli@opbg.net (R.S.); sara.cairoli@opbg.net (S.C.); 3Pediatrics Unit, Women’s and Children’s Health Department, University Hospital of Modena, 41100 Modena, Italy; fracandia92@gmail.com (F.C.); cresara90@gmail.com (S.C.); lorenzo.iughetti@unimore.it (L.I.); 4Neonatal Unit, Women’s and Children’s Department, Bufalini Hospital of Cesena, 47521 Cesena, Italy; catespa@gmail.com; 5PhD Program in Clinical and Experimental Medicine, University of Modena and Reggio Emilia, 41100 Modena, Italy; miselli.fnc@gmail.com; 6Neonatal Unit, Women’s and Children’s Department, Valduce Hospital, 22100 Como, Italy; daniele.merazzi@gmail.com; 7Neonatal Intensive Care Unit, Women’s and Children’s Department, Ca’ Foncello Hospital, 31100 Treviso, Italy; paola.lago9@gmail.com

**Keywords:** hypoxic–ischemic encephalopathy, therapeutic hypothermia, fentanyl, pharmacokinetics

## Abstract

Therapeutic hypothermia (TH) is the standard of care for newborns with moderate to severe hypoxic–ischemic encephalopathy (HIE). Discomfort and pain during treatment are common and may affect the therapeutic efficacy of TH. Opioid sedation and analgesia (SA) are generally used in clinical practice, and fentanyl is one of the most frequently administered drugs. However, although fentanyl’s pharmacokinetics (PKs) may be altered by hypothermic treatment, the PK behavior of this opioid drug in cooled newborns with HIE has been poorly investigated. The aim of this phase 1 study protocol (Trial ID: FentanylTH; EUDRACT number: 2020-000836-23) is to evaluate the fentanyl time-concentration profiles of full-term newborns with HIE who have been treated with TH. Newborns undergoing TH receive a standard fentanyl regimen (2 mcg/Kg of fentanyl as a loading dose, followed by a continuous infusion—1 mcg/kg/h—during the 72 h of TH and subsequent rewarming). Fentanyl plasma concentrations before bolus administration, at the end of the loading dose, and 24-48-72-96 h after infusion are measured. The median, maximum, and minimum plasma concentrations, together with drug clearance, are determined. This study will explore the fentanyl time-concentration profiles of cooled, full-term newborns with HIE, thereby helping to optimize the fentanyl SA dosing regimen during TH.

## 1. Introduction

Neonatal encephalopathy caused by perinatal asphyxia is one of the major causes of neonatal mortality and morbidity on a global scale [1]. Therapeutic hypothermia (TH) is effective in reducing both death and disability among full-term newborns with moderate to severe hypoxic–ischemic encephalopathy (HIE) [2,3,4,5]. Therefore, TH is now considered the standard of care for newborns with moderate to severe HIE. Nevertheless, discomfort and pain during cooling are expected, and this may affect the therapeutic efficacy of TH [2,3,4,5].

Animal studies have shown that in fetal sheep, mild hypothermia leads to a more prolonged increase in circulating cortisol levels after asphyxia, thus contributing to neuronal loss [6,7]. Furthermore, Thoresen et al. reported that cooling stress had adverse effects on asphyxiated piglets, potentially reducing the beneficial effect of hypothermia [8,9]. In fact, mild hypothermia was not protective in unsedated piglets, but the severity of hypoxic–ischemic brain injury was reduced in piglets receiving halothane or intravenous anesthesia [9].

In adults who undergo hypothermia after cardiac arrest, the administration of sedative and analgesic medications has been associated with the earlier achievement and more effective maintenance of target temperatures. Additionally, the administration of sedative–analgesic drugs decreased the occurrence of neurological complications, such as delirium or seizures [10].

In newborns, opioids are commonly used to treat pain during mechanical ventilation or painful procedures [11]. Among previous randomized controlled trials on TH, some studies have reported the use of opioids in sedation and analgesia (SA) [2,4]. An Italian survey showed that medications for SA are regularly (in 95.7% of NICU) administered during TH: fentanyl was the most frequently administered drug (84.3%), while morphine was administered in only 7% of neonatal intensive care units (NICUs) [12]. 

However, despite the need to ensure adequate analgesia and to guarantee neuroprotection, the opioid effect in asphyxiated newborns treated with TH has been poorly investigated. Alterations in drug pharmacokinetics (PKs) are expected to occur since the organs involved in drug elimination (i.e., the liver and kidneys) are subjected to frequently associated hypoxic–ischemic injuries [13,14,15,16,17]. Perinatal asphyxia can also exert an impact on cardiac output, which holds particular significance in the context of drugs with high extraction rates (such as fentanyl). This is due to the fact that metabolizing these drugs relies on optimal hepatic perfusion [18]. Furthermore, potential changes in organ physiology and drug metabolism during TH have been reported [15,16]. Another important aspect to consider is the unique pharmacologic behavior that characterizes this vulnerable population. For example, a recent clinical study examining the PKs of gentamicin in asphyxiated neonates undergoing TH showed a marked reduction in gentamicin clearance during TH and the need for extended dosing intervals [17]. Roka et al. studied morphine concentrations in cooled, asphyxiated infants and found higher levels in cooled newborns compared to normothermic patients [14]. In addition, Frymoyer et al. showed that morphine clearance was significantly lower in cooled infants with HIE compared to full-term newborns without HIE in previous studies [19]. Therefore, a deeper exploration of the PK behavior of the drugs used in the HIE population can provide further evidence-based dose recommendations.

Although fentanyl is commonly used for SA in infants affected by HIE and treated with TH, to the best of our knowledge, there are no studies that have specifically focused on fentanyl’s PKs in cooled infants. Fentanyl is a synthetic opioid with higher analgesic potency, a faster onset, and a shorter duration of action than morphine. It is frequently used in neonates because of its minimal effect on hemodynamic stability. The recommended i.v. doses of fentanyl for neonates are a loading bolus of 0.5–3 mcg/kg, followed by an infusion of 0.5–3 mcg/kg/h. Doses per kilo can be increased according to gestational age and postnatal age [20,21,22,23,24]. Depending on postnatal age and birthweight, fentanyl concentrations increased slowly after the start of therapy for both intermittent boluses and continuous infusion, and they reached a maximum concentration at 12–48 h [25,26]. Fentanyl is mainly metabolized in the liver by the isoform CYP3A4 of the microsomal cytochrome P450 superfamily, and its metabolism greatly depends on hepatic blood flow [11,12,20,21,22,25,26]. Actually, our knowledge of fentanyl PKs during hypothermia derives from studies conducted on animals and adult patients that have shown alterations in fentanyl metabolism and clearance. Experimental studies have demonstrated a 25% increase in the plasma concentration of fentanyl at a core body temperature of 32 °C [22,23,24,27]. Therefore, for newborns with HIE receiving TH, there is an increased risk of fentanyl accumulation due to an impaired drug metabolism. It is, therefore, crucial to define the optimal dosing regimen for cooled newborns with HIE. 

The aim of this phase 1 study protocol (Trial ID: FentanylTH; EUDRACT number: 2020-000836-23) is to evaluate the fentanyl time-concentration profiles of term newborns affected by HIE treated with TH.

## 2. Methods

### 2.1. Study Design and Setting

FentanylTH is a phase 1, open-label, unmasked, monocentric study on fentanyl PKs in term newborns with HIE who have been treated with TH. Patients are hospitalized at the NICU of the University Hospital of Modena, which is a level 3 NICU with 20 beds and about 3000 live births per year. The primary aim is to evaluate the fentanyl time-concentration profiles of term newborns affected by HIE and treated with TH. The secondary aim is to correlate the Neonatal Pain, Agitation, and Sedation Scale (N-PASS) [28] with fentanyl plasma concentrations.

### 2.2. Hypothesis

The TH treatment of newborns could increase fentanyl plasma levels due to reduced hepatic clearance as a consequence of impaired liver blood flow and hepatic CYP3A4 activity [15,18,27]. We hypothesize that fentanyl may accumulate and reach high plasma levels in cooled infants.

### 2.3. Inclusion Criteria

Newborns affected by moderate to severe HIE and treated with TH at the NICU of the University Hospital of Modena since 1 January 2022 are included in this phase 1 study. The selection of patients for TH is based on previously established criteria and subsequently adapted specifically for the purpose of this study: (1) gestational age ≥ 37 weeks; (2) intrapartum asphyxia, defined by at least one of the following: an Apgar score ≤ 5 at 10 min, ventilation with an endotracheal tube or a mask for at least 10 min, or metabolic acidosis within 60 min of birth (cord pH or any arterial/venous pH ≤ 7.0 or base defect ≥ 12 mmol/L); (3) neonatal encephalopathy evaluated within 1 h of birth; and (4) moderate to severe EEG polygraphic (pEEG) or amplitude-integrated EEG (aEEG) abnormalities [2,3,4,29,30,31]. This study does not include near-term newborns. This choice is due to potential differences in the fentanyl time-concentration profiles of near-term newborns with respect to full-term newborns.

The study’s other exclusion criteria are congenital malformations, chromosomal abnormalities, either suspected or confirmed metabolic disorders, sepsis or central nervous system infections, and different causes of asphyxia (i.e., sudden, unexpected postnatal collapse).

Patients are cooled to a rectal temperature of 33.5 °C for 72 h (CritiCool MTRE, Charter Kontron, Milton Keynes, UK). CritiCool functions as a control unit that constantly provides feedback on a patient’s core and surface temperatures via connected sensors. An appropriately sized CureWrap garment is wrapped around the patient. The desired temperature (33.5 °C) is set on the CritiCool device, and then the patient is cooled to 33.5 °C for 72 h. After 72 h of cooling, slow, controlled, and monitored rewarming is started, increasing the set temperature by 0.5 °C every hour. In this way, the patient achieves a 36 °C core temperature in about 5 to 6 h. If any complication occurs during rewarming (i.e., seizure), rewarming is transiently interrupted and restarted when clinically indicated [24].

Every enrolled patient is administered an indwelling double-lumen umbilical catheter with one of the two lumina dedicated to fentanyl infusion. The N-PASS algometric pain scale is routinely administered to cooled infants [28]. Twenty patients are expected to be enrolled in this study.

### 2.4. Intervention

Asphyxiated neonates who fulfill the study’s inclusion criteria are recruited. All newborns treated with TH receive SA with fentanyl since the beginning of TH (Figure 1). The administered drug is Fentanyl-Hameln, a 50 mcg/mL injectable solution (hameln pharma plus gmbh Langes Feld 13, 31789 Hameln, Germany). Before infusion, the fentanyl is diluted for both the loading dose and the continuous infusion as follows: 2 mL of a fentanyl vial (100 mcg) is diluted with 23 mL of distilled water so that 1 mL contains 4 mcg of fentanyl. Enrolled newborns receive a fentanyl loading dose of 2 mcg/kg as an intravenous bolus (in 20 min), which is immediately followed by 1 mcg/kg/h as a continuous infusion during TH and the rewarming phase. The continuous fentanyl infusion can be increased by 25–50% if discomfort or pain arises. Shivering, a 20% increase in heart rate over the baseline, facial grimaces, and N-PASS scores >3 are considered signs of discomfort or pain. During the study, no more than 60 consecutive minutes of interruption are allowed.

The following concomitant medications are allowed (according to clinical indications): inotropes (dobutamine, dopamine, milrinone, epinephrine, and norepinephrine), local anesthetics (an eutectic mixture of 2.5% lidocaine and 2.5% prilocaine and lidocaine), midazolam, phenobarbital, and phenytoin (the dose and time of administration should be recorded). Conversely, the use of the following drugs is not allowed: acetaminophen, morphine, muscle relaxants, and ketamine.

### 2.5. Pain Measurement

During the study period, the N-PASS algometric scale is used to measure pain [28]. N-PASS is a clinically relevant tool used to assess pain and sedation levels in neonates and infants. The scale encompasses five indicators: (1) crying/irritability (with a silent cry observed in intubated infants scored as a cry), (2) behavioral state, (3) facial expression, (4) extremities/tone, and (5) vital signs. The criteria are assigned values of 0, 1, or 2 (for pain/agitation) and 0, −1, or −2 (for sedation) [28]. The N-PASS pain scores range from 0 to 10 for term newborns. An N-PASS score >3 indicates moderate pain. The N-PASS score for sedation ranges from 0 to −10 (moderate sedation: an N-PASS score between −2 and −5; profound sedation: an N-PASS score between −5 and −10) [28]. In this study, the targeted level of sedation is moderate sedation (an N-PASS score between −2 and −5). The N-PASS scores for pain and sedation are recorded. 

### 2.6. Determination of Fentanyl Plasma Levels via LC-MS/MS and PK Evaluation

Fentanyl plasma levels are determined via high-performance liquid chromatography (HPLC), coupled with mass spectrometry (MS/MS). Blood samples are collected before the loading dose (T0), 30 min after the loading bolus (T after bolus), and 24-48-72-96 h after the starting bolus (T24, T48, T72, and T96, respectively). If a dosing increase occurs during the infusion, an additional blood sample is taken 30 min after the dosing adjustment. Blood samples (0.5 mL) are obtained via heel puncture and collected through a capillary tube into a microtube containing EDTA. The plasma is separated via centrifugation at 3000× *g* for 10 min and stored at −80 °C until analysis. 

The determination of fentanyl plasma levels is performed at the Division of Metabolic Diseases and Drug Biology of Bambino Gesù Children’s Hospital, IRCCS, in Rome (Italy). Briefly, 100 µL of a working solution (100 ng/mL) of deuterated fentanyl (FEN-D5), used as an internal standard (IS), is added to 50 µL of plasma or a calibration standard or quality control sample (QC). The samples are mixed via vortexing for 30 sec; thereafter, protein precipitation is carried out via the addition of 350 µL of acetonitrile (ACN). After mixing for 30 s and centrifuging at 13,000 rpm for 9 min at room temperature, 200 µL of supernatant from each tube is transferred to vials and injected into the UHPLC-MS/MS system for analysis. The stock solutions of fentanyl and FEN-D5 have a concentration of 1 mg/mL and 0.1 mg/mL, respectively. A calibration curve is prepared via serial dilutions of the fentanyl stock solutions. The dilutions are carried out using human blank plasma to obtain seven calibration standards at 0.1, 1, 10, 50, 100, 500, and 1000 ng/mL. The IS working solution is prepared at 100 ng/mL via the dilution of the corresponding stock solution with ACN. For this analysis, three QC samples are included at low (L-QC), medium (M-QC), and high (H-QC) concentrations of 2.5, 25, and 250 ng/mL, respectively. 

The liquid chromatography (LC) system consists of a UHPLC Agilent 1290 Infinity II (Agilent Technologies Santa Clara, United States). Chromatographic separation is performed in the reverse phase mode with a Kinetex^®^ 2.6 µm Polar C18 (100 × 2.1 mm) column maintained at 40 °C. The mobile phase is delivered at a flow rate of 0.5 mL/min through gradient elution, and it consists of 5 mM of ammonium formiate and 0.01% formic acid in milli-q pure water (aqueous mobile phase A) and 0.01% formic acid in methanol (organic mobile phase B). The analytical run time for each injection is 7.50 min. The initial gradient conditions start with 10% of mobile phase B. Mobile phase B is increased to 50% at 3.0 min and further increased to 95% at 4.0 min. These conditions are held for 2 min, and then the initial conditions are reconducted over 0.1 min and maintained for 1.4 min. The injection volume is 5.0 μL. The detection of fentanyl and IS (FEN D5), based on the peaks’ mass to charge (*m*/*z*) ratio, is carried out using a 6470 Mass Spectrometry system (Agilent Technologies) equipped with an ESI-JET-STREAM source operating in the positive ion (ESI+) mode. The mass spectrometric conditions are as follows: a gas temperature of 350 °C, a gas flow of 12 l/min, a sheath gas temperature of 250 °C, a sheath gas flow of 11 l/min, a 2000 V capillary, and a 50 psi nebulizer. Samples are detected in a multiple-reaction monitor (MRM) mode. The mass transitions for fentanyl are *m*/*z* 337.23 → 188.1 for the quantifier and 337.23 → 105 for the qualifier; the mass transitions for FEN-D5 are *m*/*z* 342.26 → 105. The software used to control this system and analyze the results is MassHunter.

Drug-free plasma is obtained from healthy volunteers recruited at the Blood Transfusion Center of the Bambino Gesù Children’s Hospital after obtaining informed consent, and it is used as a matrix for standard curve preparation and negative controls. Prior to use, the fentanyl and FEN-D5 stock solutions and human blank plasma are stored at −80 and −20 °C, respectively. Method validation is based on the European Medicines Agency (EMA) guidelines for bioanalytical method validation. Method validation includes the evaluation of specificity, linearity, inter- and intra-precision and accuracy, extraction recovery, the matrix effect, and stability [32].

Fentanyl time-concentration profiles are studied: the medians and ranges of fentanyl concentrations are calculated for all newborns at each time point (24, 48, 72, and 96 h). As several factors can impact the attainment and maintenance of a steady state in neonates, in this study, we are going to assess whether a fentanyl steady state is achieved or not during the observation period.

A steady state is reached when fentanyl concentrations in 2 consecutive samples are within 15% of each other without a consistent increase or decrease in the slope of the concentration timeline [25]. In the event that the steady state is reached, other pharmacokinetic parameters can be calculated: maximal plasma concentration, minimal plasma concentration, the time of peak concentrations, and the area under the plasma time-concentration curve from 0 to 72 h (AUC 0–72). Total body clearance (Cl) can be calculated by dividing the infusion rate by the concentration at the steady state [25,33].

### 2.7. Instrumental and Laboratory Examinations

Table 1 shows the assessments to be performed. Routine blood testing (including troponin, transaminases, creatinine, and urea levels) are performed. During TH and the rewarming phase, pEEG or aEEG monitoring is continued in order to document background anomalies and the occurrence of seizures during the study period. The necessity of mechanical ventilation or an increased demand for respiratory support is documented. A heart ultrasound is performed for all newborns to diagnose myocardial dysfunction. A brain ultrasound is performed at enrollment and within 7 days of life. A cerebral MRI is performed within the first two weeks of life.

### 2.8. Adverse Events

Adverse events that are ongoing in the first 4 days of life are recorded. All the adverse events noticed by the experimenter during the execution of clinical and laboratory tests on each patient are recorded and constantly monitored.

An adverse event is defined as any unfavorable and unintended sign, symptom, or disease temporarily associated with the use of a medicinal product, whether or not it is considered related to the product. Clinical conditions/illnesses before treatment with the study drug are considered adverse events only in the case of worsening after drug administration. Any alteration in laboratory exams or tests are considered adverse events only if they are related to the appearance of clinically relevant signs or symptoms and require targeted therapies. Every alteration, including the signs, symptoms, and associated diagnoses, is recorded throughout the whole study period. A serious adverse event is any untoward medical occurrence that, at any dose, results in a patient’s death, endangers the patient’s life, prolongs existing hospitalization, results in persistent or significant disability, or can damage the patient’s health and requires a medical or surgical intervention to prevent the above conditions. Any serious adverse event that occurs after informed consent has been obtained or one day after the discontinuation of the study drug must be reported. 

The neonatal adverse event severity scale (NAESS scale) is used to evaluate the severity of an adverse event [35]. The modified version of the Naranjo algorithm for newborns is adopted to assess causality. This algorithm is based on 13 items (yes/no/not applicable) that can be quantified (≥14, 7–13, 3–6, or ≤2) and can facilitate the categorization of causality (definite, probable, possible, unlikely, or not related) [36].

In newborns, fentanyl adverse events of special interest (AESI) include chest wall rigidity and glottis rigidity, respiratory depression, withdrawal syndrome, tolerance, hypotension, urinary retention, and ileus. Naloxone, a competitive opioid receptor antagonist, can be used at a dosage of 0.1 mg/kg to reverse chest wall rigidity. Due to the potential release from deep compartments (such as the muscles and adipose tissue), neonates are monitored for at least 24 h after fentanyl discontinuation. 

Any recurrent event, complication, or increased rate of an already recorded event must be registered as a “follow up” of that episode.

### 2.9. Follow-Up Phase

Adverse events that are ongoing in the first 4 days of life, as well as clinical outcomes, are assessed until hospital discharge. Moreover, the following information is collected at hospital discharge: neonatal clinical status, length of stay, duration of ventilation, and days of life when autonomous suction is achieved (if reached).

## 3. Statistical Analysis

Demographic and clinical details (race, sex, prenatal care, type of delivery, location of birth, Apgar score, severity of encephalopathy, and seizures) are summarized using descriptive statistics. Continuous variables are expressed as medians and interquartile ranges (IQRs), and categorical data are expressed as numbers (percentages). We compare categorical and continuous variables using a χ2 test, Fisher’s exact test, or Mann–Whitney test as appropriate. All *p*-values correspond to two-tailed tests of significance; *p* < 0.05 is considered significant.

## 4. Discussion

Limited dosing information is currently available on drugs that are commonly administered to HIE neonates receiving TH [37,38,39,40]. Because of alterations in drug PKs, this vulnerable population requires unique clinical pharmacologic considerations [15,16,17,18,19]. HIE infants treated with TH require SA to improve both discomfort and pain from cooling, intubation, and mechanical ventilation [39,40,41,42,43]. To date, there are no standardized dosing guidelines for SA during TH [40,41]. Actually, the European hypothermia trial is the only one out of four earlier randomized controlled trials [2,3,4,36,37] to recommend a standard dose of morphine at 100 mg/kg every 4 h or in combination with fentanyl at equipotent doses. The Italian recommendations for TH in infants with HIE suggest adopting the same SA regimen used for mechanical ventilation during normothermia [29,43]. However, the PKs and pharmacodynamic properties of opioids and their metabolites are temperature-dependent. Moreover, because of cooling and hypoxic–ischemic liver and renal injuries, fentanyl and morphine metabolism is reduced in asphyxiated infants treated with TH [14,15,16,17,18,19]. Therefore, prolonged SA during TH may lead to the toxic accumulation of sedative drugs and their metabolites, as reported in the TOBY trial [14]. Although morphine exposure is associated with adverse neurodevelopmental outcomes in preterm infants, no evidence of an adverse dose-dependent effect of morphine and fentanyl exposure on cognition, language, or motor function at 24 months of age has been reported for term infants undergoing TH [24,39]. 

Although SA is usually given during TH [4,12,14,19], firm evidence on the optimal therapeutic regimen is still lacking, and clinical practices may vary from one center to another [12,18,24,39]. 

Although neonates undoubtedly feel pain and stress during TH, the response to noxious stimuli can be difficult to interpret, whereas the severity of illness might decrease the infant’s ability to exhibit pain signals. Shivering, hypertonia, or inconsolable crying may indicate discomfort or pain during TH. Nevertheless, neonatal pain remains particularly difficult to assess [24,44,45,46].

Currently, consensus guidelines aiming to evaluate the therapeutic need for the pain relief and sedation of neonates are not yet available. Similarly, common dosing strategies to ensure an adequate therapeutic response are lacking. However, there are many different scoring tools available. The N-PASS scale [28] was developed as a clinically relevant tool to assess prolonged pain and sedation in infants, as well as acute procedural pain. The validity and reliability of N-PASS for prolonged pain have been reported [28,46,47]. The N-PASS scoring tool has been implemented with cooled newborns with HIE, and currently it is the most suitable algometric score for these patients as it assesses both pain and sedation [46]. In our study protocol, we are adopting the N-PASS score to titrate fentanyl infusion administration. To obtain adequate SA, the study’s fentanyl dosage will be guided by clinical behavioral or physiological pain indicators, as specified by the N-PASS scale. 

An excess of SA can be harmful to newborns undergoing TH. Respiratory depression is the most severe opioid side effect that can occur in spontaneously breathing infants during TH [24]. A reduction in the respiratory drive can result from both encephalopathy and opioid administration. In fact, a reduction in the hepatic metabolism can lead to drug accumulation during TH [14,19,20]. The current practice guidelines recommend invasive ventilation for pediatric patients during TH [48]; however, recommendations for the respiratory management of cooled newborns are lacking, and clinical approaches differ among centers. Data retrieved from the Vermont Oxford Network Neonatal Encephalopathy Registry (from 2006 to 2010) show that 64% of eligible infants receive mechanical ventilation, although NICUs’ cooling protocols for cases of mild encephalopathy have recently been reported to involve a lower rate (30%) of mechanical ventilation [42,49]. Orsini et al. reported that fentanyl may prolong ventilatory support until day 3 of life when doses higher than those suggested in the present study are given [43]. 

In addition, it is still uncertain whether fentanyl withdrawal syndrome may occur after TH. Withdrawal syndrome is related to both the total administered dose and the duration of treatment, and it may occur in most patients treated with a cumulative dose of 2.5 mg/kg for more than 9 days [21,50]. However, when fentanyl is given for less than 72 h to normothermic neonates, no gradual discontinuation is required [11]. Accordingly, in the present study, fentanyl is immediately discontinued at the end of the rewarming phase. 

In conclusion, adequate and safe SA is not easy to achieve during TH and, therefore, requires close and continuous monitoring. Our FentanylTH study aims to explore the fentanyl time-concentration profiles of cooled HIE newborns, which will be systematically evaluated using an algometric scale, thereby helping to define the optimal fentanyl dosing regimen for this vulnerable population.

## Figures and Tables

**Figure 1 biomedicines-11-02395-f001:**
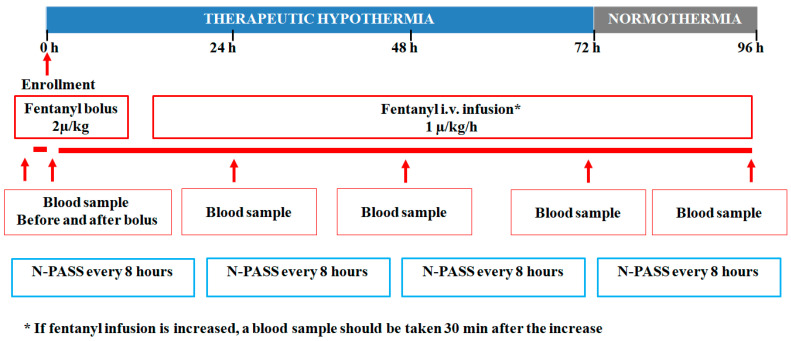
FentanylTH protocol timeline schedule.

**Table 1 biomedicines-11-02395-t001:** Assessment schedule.

	Pre-Recruitment	Study Phase				Follow-Up	
		Day 1	Day 2	Day 3	Day 4	Hospital Discharge	24 Months
Informed consent	X						
Prenatal history and demographics	X						
Physical exam	X						
Heart rate every 4–6 h	X	X	X	X	X		
Hourly rectal temperature	X	X	X	X	X		
Neurologic exam	X	X	X	X	X	X	X
aEEG/pEEG	X	X	X	X	X		
Head ultrasound	X	X					
N-PASS scale ^1^	X	X	X	X	X		
Record painful/stressful procedures	X	X	X	X	X		
Record concomitant drugs and dosage	X	X	X	X	X		
Record mechanical ventilation and the need for increased respiratory support	X	X	X	X			
Record minimal enteral feeding tolerance	X	X	X	X	X		
Diuresis	X	X	X	X	X		
Record renal dysfunction ^2^	X	X					
Hepatic dysfunction ^3^	X	X					
Troponin	X	X					
Record blood pressure, drugs for cardiovascular support, and vasoactive inotropic score (VIS) ^4^	X	X	X	X	X		
Heart ultrasound	X	X					
Record apnea/need for resuscitation	X	X	X	X	X		
Record length of mechanical ventilation, assess survival and length of stay						X	
Developmental assessment (Griffiths or Bayley tests)							X
Cerebral MRI						X	X

^1^ Recorded every 8 h. ^2^ Creatinine > 1.5 mg/dl and diuresis < 1 mL/Kg/h. ^3^ Liver enzyme > twice normal value. ^4^ [34].

## Data Availability

Anonymous data are available and can be shared upon request.

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
