# Peer review of "Continuous Fentanyl Infusion in Newborns with Hypoxic–Ischemic Encephalopathy Treated with Therapeutic Hypothermia: Background, Aims, and Study Protocol for Time-Concentration Profiles"

_biomedicines, 2023, doi:10.3390/biomedicines11092395_

Round 1
Reviewer 1 Report
i have added an annotated and commented pdf file. I hereby have tried to provide additional input to try to further improve the reporting.
In essence, this work is important, but the description on the methods needs additional considerations, and i disagree on the a priori statement of no data sharing (cfr ethics)

Author Response
Dear Reviewer,
Thank you for your pertinent comments, helping us to improve the paper, given Your evident expertise in the field. You find enclosed in the table responses to Your comment, while text changes are yellow highlighted.
The revised version of the paper is attached.
Best regards
Licia Lugli

Reviewer 2 Report
In the paper entitled "Continuous fentanyl infusion in newborns with hypoxic ischemic encephalopathy treated with therapeutic hypothermia: background, aims and pharmacokinetics study protocol" the authors studied fentanyl pharmacokinetics in cooled via hypothermia full-term newborn. This paper is well written and definitely will be very helpful for clinicians. In the method section, the authors didn't describe standard protocol for the hypothermia procedures which might vary between different countries. The authors need to describe the hypothermia procedure protocol used in this study.
Author Response
Dear Reviewer,
thank You for Your comment. We have modified the paper according to your suggestion. We have described the protocol for systemic hypothermia that we are adopting in the study (changes are highlighted in yellow on page 5)
Best regards
Licia Lugli
Round 2
Reviewer 1 Report
the suggestions have been handled well, sorry for the pmid typo, but the 'idea' is in the revised version